# Role of AHR Ligands in Skin Homeostasis and Cutaneous Inflammation

**DOI:** 10.3390/cells10113176

**Published:** 2021-11-15

**Authors:** Nieves Fernández-Gallego, Francisco Sánchez-Madrid, Danay Cibrian

**Affiliations:** 1Immunology Service, Hospital Universitario de la Princesa, Universidad Autónoma de Madrid (UAM), Instituto de Investigación Sanitaria del Hospital Universitario de La Princesa (IIS-IP), 28006 Madrid, Spain; nieves.fernandez-gallego@uam.es; 2Vascular Pathophysiology Area, Centro Nacional de Investigaciones Cardiovasculares (CNIC), 28029 Madrid, Spain; 3CIBER de Enfermedades Cardiovasculares, Instituto de Salud Carlos III, 28029 Madrid, Spain

**Keywords:** aryl hydrocarbon receptor, cutaneous inflammation, immune and inflammatory responses, AHR signaling pathways, AHR endogenous and exogenous ligands

## Abstract

Aryl hydrocarbon receptor (AHR) is an important regulator of skin barrier function. It also controls immune-mediated skin responses. The AHR modulates various physiological functions by acting as a sensor that mediates environment–cell interactions, particularly during immune and inflammatory responses. Diverse experimental systems have been used to assess the AHR’s role in skin inflammation, including in vitro assays of keratinocyte stimulation and murine models of psoriasis and atopic dermatitis. Similar approaches have addressed the role of AHR ligands, e.g., TCDD, FICZ, and microbiota-derived metabolites, in skin homeostasis and pathology. Tapinarof is a novel AHR-modulating agent that inhibits skin inflammation and enhances skin barrier function. The topical application of tapinarof is being evaluated in clinical trials to treat psoriasis and atopic dermatitis. In the present review, we summarize the effects of natural and synthetic AHR ligands in keratinocytes and inflammatory cells, and their relevance in normal skin homeostasis and cutaneous inflammatory diseases.

## 1. Introduction

The skin is the first line of defense against a multitude of pathogens and environmental threats. The outermost layer of the epidermis, the stratum corneum, is a cornified envelope that acts as a physical barrier, which is generated by keratinocyte terminal differentiation and death. The process of terminal keratinocyte differentiation involves the upregulated expression of specific proteins such as involucrin (IVL), loricrin (LOR), and filaggrin (FLG). Any compromise of the integrity of the skin barrier function can lead to dryness, itchiness, or flakiness (or all three). It also participates in pathogenic conditions such as atopic dermatitis (AD) and psoriasis (PS) [1].

AD and PS are frequent inflammatory cutaneous disorders, in which deregulation of immune cells is accompanied by alterations in keratinocyte differentiation, proliferation and overall barrier function [2,3]. AD features a Th2-polarized immune response with increased interleukin (IL)-4 and IL-13 levels [3]. Psoriatic lesions are characterized by upregulation of tumor necrosis factor-α (TNF)-α and a Th17 response with numerous IL-17-secreting cells, activated by elevated levels of IL-23 [2]. Although they are treatable using biological therapies aimed at blocking IL-4/IL-13 in AD [4], and TNF-α/IL-23/IL17 in PS [5,6], both conditions remain uncured, and their frequent relapse episodes deteriorate the quality of life of the patients. In addition, some major autoimmune disorders are comorbidities for AD and PS [2,3].

Abnormal skin barrier integrity is particularly relevant for the onset of AD and is associated with the reduced production of terminal differentiation molecules such as FLG [7,8]. Transepidermal water loss (TEWL) is a non-invasive in vivo measurement of water loss across the stratum corneum, which is raised in AD patients, at both lesional and non-lesional skin sites [9,10]. A defect in skin barrier function may facilitate allergen entry and immune priming. Skin barrier dysfunction also increases colonization by *Staphylococcus aureus*, which further exacerbates Th2 polarization [11]. Moreover, several pro-inflammatory cytokines such as IL-4, IL-13, TNF-α, IL-17, and IL-22 alter the expression of proteins involved in skin barrier function [12].

Aryl hydrocarbon receptor (AHR), also termed dioxin receptor, is a ligand-dependent transcription factor that can be activated by a plethora of exogenous and endogenous environmental agents, including polycyclic aromatic hydrocarbons (PAHs) and halogenated aromatic hydrocarbons (HAHs, or dioxins), as well as metabolites derived from L-tryptophan (L-Trp) catabolism [13]. Every skin cell type expresses AHR, including keratinocytes, sebocytes, fibroblasts, melanocytes, endothelial cells, Langerhans cells, and lymphocytes [14]. Activation of AHR upregulates the expression of barrier-related proteins and accelerates terminal keratinocyte differentiation [15,16,17]. Hence, skin homeostasis, as well as cutaneous pathological processes such as AD and PS, can be modulated by specific ligand-dependent activation of AHR. In this review, we summarize the diverse roles of exogenous and endogenous AHR ligands in skin homeostasis, as well as in the treatment of AD and PS (Table 1).

### 1.1. AHR as a Sensor of Environmental Cues

The AHR was first described as a cytosolic receptor that binds PAHs, e.g., 3-methylcholanthrene and HAHs, e.g., 2,3,7,8-tetrachlorodibenzo-p-dioxin (TCDD). Ligand binding to AHR induces the synthesis of cytochrome P450 (CyPs) enzymes, which are involved in the metabolism of xenobiotic compounds and the generation of reactive oxygen species (ROS) [16,18]. Beyond its role as a regulator of xenobiotic metabolism, the AHR is also an important modulator of various physiological functions associated with the presence of endogenous, non-xenobiotic ligands, including host defense. Many molecules have been identified as AHR ligands, suggesting that AHR contains a relatively promiscuous ligand-binding site [19]. AHR ligands trigger a myriad of functions, and their ultimate effects depend on several factors, including their concentration, the cell type and cellular context, or their interaction with antagonistic molecular pathways, such as hypoxia-inducible factor (HIF)-1α and EGFR in T cells and keratinocytes, respectively [15,20].

Although the AHR is evolutionarily conserved across metazoan phyla, it is remarkable that the AHR exhibits interspecies differences or even among strains [21,22]. Comparison between mouse and human AHR revealed around 86% amino acid sequence homology in the N-terminal half of the receptor; whereas, the C-terminal half exhibits only 58% identity [23]. Most of the non-conserved changes of the AHR are found in the transcriptional activation domain (TAD), resulting in differential protein–protein interactions with other coactivators, corepressors, or nuclear receptors, which may result in differential gene expression regulation [24]. Indeed, studies using primary hepatocytes derived from humans, mice, and humanized mice, which specifically express human AHR, have revealed that the human and mouse AHR regulate different genes subsets involved in several biological pathways [25,26]. Altogether, these results reflect the complexity of AHR function and the difficulties of translating studies from experimental animals to human physiology.

### 1.2. AHR Signaling Pathways

AHR controls biological processes through genomic and non-genomic signaling events. Genomic signaling involves a canonical and a non-canonical pathway, with the former being the best characterized.

In the canonical pathway (Figure 1), the AHR functions as a ligand-activated transcription factor that directly regulates the expression of a wide range of target genes, named the AHR gene battery—such as CYP1A, CYP1A2, and CYP1B1 enzymes of the CyP family [27,28,29,30]—AHR repressor (AHRR) [31], and TCDD Inducible Poly (ADP-Ribose) Polymerase (TIPARP) [32,33]. Regulatory regions located in the upstream of AHR gene battery contain a DNA consensus sequence (5ʹ-TNGCGTG-3ʹ) known as the xenobiotic responsive element, XRE (also known as the dioxin responsive element (DRE) or the AHR-responsive element (AHRE)), which acts as a transcriptional enhancer and an AHR binding site [27,28,29,30].

Structurally, AHR belongs to the family of basic helix–loop–helix or periodic circadian protein–AHR nuclear translocator single-minded protein (bHLH/Per-ARNT-SIM or PAS) domain-containing transcription factors [34,35,36]. The bHLH motif and two PAS (A and B) domains are located in the N-terminal region [37]. The bHLH motif is involved in DNA binding and dimerization of proteins and the PAS domains also participate in protein–protein interactions [37,38]. Additionally, the AHR contains a TADs in the C-terminal region [39,40].

In the absence of ligands, the AHR is confined in the cytosol that is associated with diverse chaperones, including a dimer of 90 kDa heat shock protein (HSP90), the co-chaperones p23, the AHR-interacting protein (AIP) (also known as ARA9 or X-Associated Protein-2 (XAP-2)), and the protein kinase Src (Figure 1) [41,42,43,44]. This chaperone complex has multiple roles towards maintaining a functional AHR as follows: it is involved in the folding and stabilization of AHR protein, it ensures its cytoplasmic retention while maintaining it in a high ligand-binding affinity conformation, and it protects AHR from ubiquitylation-mediated proteasomal degradation [41,45,46,47].

Upon agonist binding, the AHR changes its conformation, translocates to the nucleus, dissociates from its chaperone complex, and forms a heterodimer with a constitutively expressed nuclear factor knowns as an AHR nuclear translocator (ARNT) or as HIF-1β (Figure 1) [44,46,48]. The AHR–ARNT heterodimeric complex is required for AHR–DNA binding and transcriptional function [40,49]. The AHR–ARNT–DNA complex then recruits the following: coactivators CBP/p300, SP1, NCOA 1-3, and RIP140; kinases IKKa, MSK1, and MSK2; components of the ATP-dependent chromatin remodeling complexes such as BRG-1 and *p*-TEFβ; and RNA initiation factors required for RNA polymerase II, which increase promoter accessibility and initiate transcription of the target genes [40,50,51,52,53].

AHR can also control the expression of genes that do not harbor XREs by interacting with additional transcription factors that direct its recruitment to target DNA sequences, different from canonical XREs. Non-canonical AHR partners include the estrogen receptor (ESR), the retinoic acid receptor (RAR), the retinoblastoma protein (RB), Krüppel-like factor 6 (KLF6), nuclear factor erythroid 2-related factor 2 (Nrf2), musculoaponeurotic fibrosarcoma (c-Maf), and nuclear factor-κB (NF-κB) [54,55,56,57,58,59].

AHR signaling is regulated at three levels, as follows: ligand metabolism by CyP enzymes, AHR–ARNT complex disruption by AHRR, and proteasomal degradation of the AHR. These mechanisms limit AHR’s signaling and protect cells from prolonged exposure to high concentrations of agonists.

Many AHR agonists (e.g., 6-formylindolo [3,2-b]carbazole or FICZ) are substrates for CyP enzymes downstream of the AHR, mainly CYP1A1, and are rapidly metabolized, thus generating only transient effects. In contrast, many xenobiotic ligands (e.g., TCDD) are stable and resistant to degradation. As a consequence, xenobiotic ligands have extended half-lives, driving prolonged AHR activation that has to be counteracted by additional control mechanisms [60,61].

Furthermore, the AHRR inhibits AHR signal transduction in two ways. First, the AHRR competes with the AHR for interaction with ARNT and XRE binding which, in turn, decreases gene expression [31,62]. Second, the AHRR has activity as a transcriptional repressor, recruiting co-repressors such as ANKLA2, HDAC4, and HDAC5, when the AHRR–ARNT–DNA complex is formed [63]. AHR’s transcriptional activity is affected by HIF1α interaction with the ARNT, which is independent of AHRR-mediated inhibition [20]. When the AHR–ARNT complex is disassembled from DNA, the AHR is exported from the nucleus and subjected to proteasomal degradation [64,65].

Several AHR-mediated non-genomic pathways, independent of DNA binding, have been identified recently. AHR ligand activation also produces an increase in intracellular Ca^2+^ concentration [66], interacts with E3 ubiquitin ligases (promoting the proteasomal degradation of target proteins) [67], or triggers phosphorylation cascades, driven by Src kinases upon its release from the AHR–chaperone cytoplasmic complex [42,43].

## 2. Role of AHR Function in Skin Immune System

The AHR plays a relevant role in many immune and inflammatory processes, such as multiple sclerosis, rheumatoid arthritis, asthma, inflammatory bowel disease (IBD), chloracne, AD, and PS [12,68,69,70,71,72]. The AHR is a critical regulator of the balance between proinflammatory or autoimmune Th17 cells and both immunosuppressive or tolerogenic Treg and T regulatory type 1 (Tr1) cell populations, which is determined by the cellular microenvironment and the presence of specific ligands (Figure 2). TCDD-induced AHR activation promotes transactivation of FoxP3 gene in vitro, and expansion of the CD4+CD25+FoxP3+ Treg-cell compartment in vivo. Moreover, AHR activation leads to epigenetic changes in the FoxP3 locus, and the expression of additional transcription factors required for the induction of functional FoxP3+ Treg cells, such as decapentaplegic homolog (Smad)1 and Aiolos [73,74,75] (Figure 2). Besides, the AHR synergizes with c-Maf and transactivates IL-10 and IL-21 in Tr1 cells [20,56,74], and facilitates RORγt-mediated IL-22 transcription in Th17 cells [76,77]. Moreover, the AHR is essential for IL-22 secretion by innate lymphocytes, including γδ T cells and innate lymphoid cells (ILC) 3 populations [77,78,79].

Interestingly, different ligands trigger the interaction of the AHR with different transcriptional partners, leading to AHR recruitment to different target DNA sequences, thereby inducing different biological responses [75,80]. For example, FICZ promotes in vivo Th17 differentiation and exacerbates experimental autoimmune encephalomyelitis (EAE), while TCDD treatment increases the pool of Treg and promotes IL-10 secretion by Tr1 cells, ameliorating disease progression [75]. However, the immunosuppressive effect of TCDD depends on the timing of administration in the EAE model, and both ligands—TCDD and FICZ—can upregulate the Th17 program in vitro, with the magnitude of response depending on AHR affinity [80]. Overall, these compelling data demonstrate that AHR activation can be modulated by multiple factors, and it can play differential roles, even in the same pathology.

Regarding the skin immune system, the AHR exerts a major role in several immune cell populations. Murine skin harbors several populations of residents and recruited γδ T cells that play essential roles in the development of PS and AD [81]. Dermal IL-17-secreting γδ T cells, as well as epidermal dendritic γδ T cells, express AHR [82]. Using the model of IL-23-induced skin inflammation, we demonstrated that CD69 expression controls AHR-mediated IL-22 expression in Th17 and γδ T cells [78]. CD69 regulates L-Trp uptake by the amino acid transporter L-type amino acid transporter 1 (LAT1), thus controlling the intracellular pool of AHR ligands, such as FICZ, in T cells. Moreover, the chemical inhibitor of AHR (CH-223191) was effective in the control of skin inflammation induced by intradermal administration of IL-23, blocking the secretion of IL-22 by Th17 and dermal γδ T cells [78]. The inhibition of the LAT1 amino acid transporter effectively blocks IL-17 and IL-22 secretion by Th17 and γδ T cells, thus preventing imiquimod (IMQ) and IL-23-induced skin inflammation [83]. LAT1 also mediates the L-Kyn efflux or influx in the blood barrier and immune cells, thus playing a major role in the regulation of AHR activation [84,85] (Figure 1).

In addition, the expression of AHR is essential for dendritic epidermal γδ T cell maintenance in mouse skin [86] and also contributes to the persistence of skin resident memory T cells (TRM) [87]. TRM cells can be CD8+ T cells or CD4+ T cells, which enter the epidermis and dermis, respectively, during infection or inflammation, and become long-lived tissue-resident populations [88]. These memory T cells are distinct from circulating effector memory and central memory populations found in circulation. TRM cells provide the earliest response to secondary challenges, playing a critical role in skin defense [88]. However, aberrant TRM cell activation contributes to the chronicity of skin inflammatory diseases such as AD and PS [88]. TRM cells remain in resolved PS lesions, secreting IL-17A and IL-22 that may cause relapse [89]. Repeated exposure to AD triggering factors also induces TRM cells, which secrete multiple cytokines in addition to Th2 response, including IL-17 and IL-22, and play a role in the recurrence and chronicity of AD [90]. Despite the relevance of AHR in the expression of IL-17 and IL-22 cytokines, the role of different AHR ligands or AHR inhibition in the control of TRM expansion and function has not been assessed in PS or AD. 

The AHR is also a critical regulator of Langerhans cells activation and function. Mice with specific deletion of AHR in langerin-expressing cells show reduced number and activation of Langerhans cells while enhancing Th2 and Tr1 responses upon epicutaneous protein sensitization [91]. AHR affects the balance between the inflammatory M1 and anti-inflammatory M2 phenotypes, increasing the secretion of proinflammatory cytokines [92]. The specific ligands involved in AHR regulation in dendritic cells (DCs) and macro-phages populations in the skin, during homeostasis and inflammation, are not identified yet.

### AHR Function in the Epidermis

The AHR is involved in many aspects of skin physiology, such as detoxification, cellular homeostasis, skin pigmentation, and skin immunity [14]. Human keratinocytes express high levels of AHR in homeostasis, but its expression is increased in inflammatory conditions, such as PS and AD [93,94]. Both AHR and ARNT colocalize in the nuclei of keratinocytes at the lower epidermis of psoriatic lesions, suggesting activation of the AHR pathway [93]. AHR upregulates the production of skin barrier proteins in vivo and in vitro, accelerating epidermal terminal differentiation of keratinocytes [95,96,97]. AHR triggers the expression of genes of the epidermal differentiation complex (EDC), which includes the cornified envelope precursor gene family, the S100A proteins, and the fused gene family. Cornified envelope precursor gene family codes for barrier-related molecules, such as LOR, IVL, late cornified envelope protein genes (LCEs), and small proline-rich protein genes (SPRRs). Most S100A proteins, such as S100A7 (psoriasin), exert antimicrobial and proinflammatory effects, especially in PS. The fused gene family includes FLG, FLG2, and hornerin (HRNR) genes, among others [12,17,97,98,99]. AHR upregulates the expression of OVO-like 1 (OVOL1) transcription factor, promoting its cytoplasm-to-nucleus translocation [99,100]. Both FLG and LOR are upregulated by the AHR–OVOL1 axis; whereas, IVL upregulation by AHR is independent of OVOL1 [98,100]. The generation of ROS is involved in the AHR’s regulation of epidermal terminal differentiation [101].

Mice with full genetic deletion of *Ahr,* [102] as well as mice expressing constitutively active AHR in keratinocytes [103], show alterations in the epidermis and cutaneous inflammation that resemble AD, suggesting the relevance of fine-tuning the AHR pathway in the skin. Severe defects in desquamation and epidermal barrier function are also observed in mice with targeted ablation of *Arnt* in keratinocytes [104]. Mice with full genetic deletion of *Ahr* also show an exacerbated form of PS, induced by IMQ, which could be reproduced by the specific genetic deletion of *Ahr* in keratinocytes, but not in immune cells, indicating its major role in curbing epidermal differentiation in psoriatic skin [105]. Interestingly, AHR protein expression and activation are downregulated in psoriatic microvascular endothelial cells, and specific deletion of *Ahr* in the endothelial compartment exacerbates skin inflammation and neutrophil recruitment in the IMQ and IL-23-induced PS models [106]. Consistently, *Ahr*-deficient endothelial cells display increased ICAM-1 expression in vivo and in vitro, which likely facilitates neutrophil recruitment to the skin [106].

The skin microbiome of *Ahr*-deficient mice is more variable and complex than that of *Ahr*-sufficient mice, reflecting difficulties in controlling stable skin microflora [107]. Enhanced susceptibility to *S. aureus* infection and AD, induced by repeated epicutaneous sensitization of tape-stripped skin with ovalbumin, was observed in mice with specific deletion of *Ahr* in keratinocytes [108]. These data demonstrate that specific alterations of AHR function in keratinocytes can lead to enhanced barrier damage and facilitate bacterial entry, thus triggering AD.

On the other hand, a mouse line whose keratinocytes express a constitutively active form of AHR develops AD-like phenotypes, exhibiting frequent scratching and increased production of Th2-type cytokines by splenic lymphocytes [103]. Using this animal model, it was demonstrated that air pollutants can induce AHR-mediated expression of the gene encoding artemin protein, an important pruritogenic factor that is highly increased in the skin of AD patients [109]. Hence, dysregulated AHR signaling in the skin can induce allergic inflammation by inducing skin microbiota dysbiosis or sensing air pollutants.

## 3. AHR Ligands in Skin Homeostasis and Inflammation

AHR ligands can be classified into two major categories: (i) natural ligands, generated in biological systems that may have endogenous (host or microbiome) or exogenous (dietary intake or microbiome) origin; (ii) synthetic ligands, such as xenobiotic or pharmaceutical agents.

### 3.1. Endogenous L-Tryptophan-Derived AHR Ligands

The essential amino acid, L-Trp, is a precursor to an important number of metabolites with AHR-inducing activities (Table 1 and Figure 3). Besides its essential role as a building block in protein synthesis, L-Trp acts as a precursor in four metabolic pathways in mammalian cells, associated microbiota, and in plants, as follows: L-kynurenine (L-Kyn), serotonin, indolic, and tryptamine [110]. These distinct pathways compete for the pool of available L-Trp and, specifically, convert L-Trp into L-Kyn, serotonin, indoles, and tryptamine, respectively. About 90–95% of ingested L-Trp is converted into L-Kyn and downstream metabolites, and around 4–6% of L-Trp is metabolized into indolic compounds, as a result of various physiochemical and biological processes in which the gut microbiota is critical. Only 1–2% of L-Trp is metabolized into serotonin (5-hydroxytryptamine) and about 1% or less goes towards the production of tryptamine, which can be initiated by either the host or the microbiota [110].

The L-Kyn pathway mainly occurs in the liver, but also in some extrahepatic tissues, e.g., the skin. AHR regulates the expression of the first two, rate-limiting enzymes of the L-Kyn pathway, tryptophan 2,3-dioxygenase (TDO), and indoleamine 2,3-dioxygenase (IDO), as well as the downstream enzymes kynureninase (KYNU) and kynurenine 3-monooxygenase (Figure 3). TDO is predominantly expressed in the liver, mainly in response to hormone signaling, controlling the levels of circulating L-Trp available for the rest of the aforementioned metabolic pathways. There are two isoforms of IDO enzymes in mice and humans, IDO1 and IDO2, which possess overlapping and distinct immune-regulatory functions. IDO1 is widely expressed; whereas, IDO2 is expressed only in the liver, kidney, and DCs and B cells in the immune system [111]. During inflammation, TDO levels are reduced and IDO levels are increased, producing L-Kyn and its downstream metabolites that exert important functions in the immune system in inflammatory, infectious, and neoplastic disorders. The presence of pathogenic and inflammatory stimuli such as LPS, CpG, and IFNγ upregulate mainly IDO1 expression, through AHR signaling in different cell types, including monocytes, macrophages, DCs, and epithelial cells [112,113,114]. In addition to L-Kyn, the IDO pathway also triggers the generation of L-Kyn-derived endogenous AHR ligands, like kynurenic acid (KynA), xanthurenic acid, or cinnabarinic acid (Figure 3), which in turn amplifies AHR signaling in the local environment, conforming a positive feedback loop [112,115,116,117,118,119,120]. IDO activity results in a reduction of free L-Trp levels, limiting microorganism growth and inhibiting T cell proliferation [114,121]. Moreover, L-Kyn induces AHR-mediated upregulation of FoxP3+, driving Treg cell differentiation [122]. These immunosuppressive functions of L-Trp-derived L-Kyn metabolite and AHR underlie the regulation of autoimmunity and resolution of inflammation in several contexts [122,123,124,125], but its efficacy in AD and PS is not clear.

The intra-lesional injection of IDO1-expressing fibroblasts in IMQ-induced psoriasis-like dermatitis significantly improves the clinical lesional appearance and reduces the infiltration of IL-17 and IL-23 by lymphocytes and DCs, respectively [126]. Accumulating evidence indicates that IDO2 acts as a pro-inflammatory mediator of autoimmunity [127]. However, when IMQ-induced, PS-like dermatitis was assessed in *Ido2*-deficient mice, skin erythema, scaling, thickness, and levels of TNF-α, IL-23p19, and IL-17A appeared increased [128] (Table 1). These data suggest that both IDO isoforms activation could control PS. The analysis of the Kyn-to-Trp ratio in serum samples indicates higher IDO activity in patients with PS than in healthy controls [129]. Although myeloid DCs from patients with PS express higher levels of IDO1 than those from healthy controls, these cells fail to upregulate IDO in response to a combination of TNF-α, IL-1β, and IL-6. The defective expression of IDO1 correlates with PASI in these patients [129]. The expression of the IDO1 enzyme is mainly upregulated in cancer cells; whereas, the levels of the KYNU enzyme are preferentially upregulated compared to IDO1 in inflammatory diseases [130]. The expression of KYNU is upregulated in keratinocytes and immune infiltrating cells in psoriatic lesions, but not in normal skin [130,131]. IL-17 and TNF-α synergistically enhance KYNU expression in keratinocytes [132]. These data indicate that L-Kyn is further degraded by KYNU in PS, thus avoiding its anti-inflammatory role. In addition, L-Trp metabolites downstream of KYNU can act as proinflammatory mediators, upregulating several cytokines, chemokines, and cell adhesion molecules [130]. KYNU expression positively correlates with disease severity and inflammation and is reduced upon successful treatment of PS or AD [130]. Recently, KYNU chemical inhibition was shown to effectively alleviate the pathological phenotypes of IMQ-induced PS [130]. Importantly, KYNU knockdown significantly blunted the induction of the inflammatory factors IL-1β, IL-6, IL-8 by keratinocytes, but whether AHR was involved in this effect was not assessed [131]. In addition, 3-hydroxy-L-kynurenamine (3-HKA), a biogenic amine produced by an alternative pathway of tryptophan metabolism (Figure 3), was protective in an experimental mouse model of psoriasis. In vitro, 3-HKA inhibited the IFN-γ-mediated STAT1/NF-κΒ pathway in both mouse and human DCs and decreased the release of pro-inflammatory chemokines and cytokines, most notably TNF, IL-6, and IL12p70 [133]. On the other hand, KynA negatively modulates in vitro the expression of IL-23 and IL-17 by DCs and CD4+ cells, respectively [134]. However, the in vivo effect of KynA and its dependency on AHR has not been explored in PS or AD models.

Moreover, several reports indicated that topical administration of L-Kyn or its derivatives, mainly KynA, efficiently attenuate fibrotic responses in vivo in wound healing models [135,136]. L-Kyn and KynA exert their AHR-mediated anti-scarring effects by the upregulation of matrix metalloproteinase (MMP)1 and MMP3 and the suppression of type-I collagen and fibronectin expression, directly on dermal fibroblasts [135,137,138]. FICZ also inhibits collagen production and promotes collagen degradation by the AHR-dependent upregulation of MMP1 in human dermal fibroblasts [139,140]. However, the immunoregulatory properties of FICZ, L-Kyn, and KynA, as well as its direct effect on keratinocytes, may provide additional mechanisms that antagonize the development of fibrosis in vivo. A novel topical treatment for keloid scars, based on KynA delivery, is under clinical trial [141,142]. Importantly, the relevance of dermal fibroblast activation and collagen deposition in PS and AD has remained mostly unexplored. However, it seems reasonable to at least test this novel anti-scarring drug, based on KynA delivery, in the development of PS and AD experimental models.

Serotonin and tryptamine pathways are also L-Trp degradation pathways that produce AHR agonists (Figure 3). Serotonin or 5-hydroxytryptamine (5-HT) is a monoamine neurotransmitter with an indolamine structure, derived from L-Trp through two enzymatic steps. The first rate-limiting reaction of the 5-HT pathway produces 5-hydroxytryptophan (5 (OH)Trp), another AHR agonist [143]. Treatment with 5 (OH)Trp inhibits IMQ-induced psoriasiform dermatitis in mice and control activation in keratinocytes and splenocytes [144] (Table 1). The serum levels of 5-HT are elevated in psoriatic patients with anxiety, correlate with disease severity, and decrease after treatment [145,146]. The expression of 5-HT and its receptors in psoriatic skin lesions is upregulated compared with normal skin, which might facilitate the development of PS by promoting the proliferation of keratinocytes and acting as an inflammatory mediator [147]. Moreover, 5-HT expression in the skin is significantly higher in patients with eczema [148], contact allergy [149], and AD [150]. Imbalanced 5-HT expression significantly correlates with the severity and extent of disease and the appearance of anxiety or depression (or both). Expression of the serotonin transporter (SERT) on inflammatory cells (e.g., DCs) in the psoriatic epidermis is also increased, and there is a positive correlation between the PS severity and the number of SERT-positive DCs [151,152]. While the role of the 5-HT system in skin inflammation has been described, further studies are required to evaluate the relevance of AHR activation in these pathways.

Tryptamine is an indolamine, synthesized by the enzymatic decarboxylation of L-Trp (1% total dietary L-Trp). In mammals, tryptamine acts as an endogenous neurotransmitter and is an AHR agonist and a CYP1A1 substrate [153]. Furthermore, tryptamine is a precursor of downstream AHR agonists including indole-3-acetylaldehyde (IAAld) and indole-3-acetic acid (IAA) [154], which are produced by host and microbiota metabolism or obtained from dietary plants (Figure 3). The potential benefits of indole derivatives are highlighted by disease amelioration in animal models of colitis and EAE [155,156,157,158,159,160]. IAA has been tested as an active agent in a novel form of photodynamic therapy used in the treatment of acne vulgaris [161], seborrheic dermatitis [162], and multiple actinic keratosis [163]. However, the role of tryptamine or its metabolism in PS or AD has yet to be explored.

### 3.2. Endogenous Ligands Derived from Photo-Oxidation of L-Trp

L-Trp photo-oxidation triggers several photochemical products that competitively bind to AHR, upregulating CYP1A1 expression [164]. These include IAA, IAAld, 1-(1H-indol-3-yl)-9H-pyrido [3,4-b]indole [165], and 6-formylindolo [3,2-b]carbazole (FICZ) [166], which has structural similarities to the potent indole ligand indolo (3,2-b)carbazole (ICZ) [167]. FICZ is an important physiological endogenous AHR agonist, with high binding affinity, similar to that of TCDD [167]. Unlike TCDD, FICZ is rapidly metabolized by CYP1A1, thus creating an important negative feedback loop [166,167]. FICZ may be formed intracellularly in skin cells by ultraviolet (UV) irradiation-induced L-Trp oxidation and has local effects [168]. However, systemic effects are likely, because FICZ-derived sulfate conjugates have been detected in human urine [169]. FICZ may be formed via other pathways distinct from L-Trp photolysis by UV light. Light-independent pathways include L-Trp oxidation by intracellular oxidants like hydrogen peroxide (H2O2) [170]. Alternatively, enzymatic deamination of tryptamine yields IAAld, the precursor of FICZ (Figure 3).

Systemic administration of FICZ in the IMQ model results in attenuated psoriasiform skin inflammation, with reduced expression of the proinflammatory mediators IL-17 and IL-22 [105] (Table 1). Recently, two new synthetic AHR, ligands structurally related to the natural agonists FICZ, NPD-0614-13, and NPD-0614-24, were assessed on two different three-dimensional models of psoriasis—a reconstructed human epidermal equivalent and a full-thickness reconstructed skin—which represents a more complex system, due to the presence of psoriatic fibroblasts [171]. NPD-0614-13 and NPD-0614-24 counteracted the altered proliferation of human primary keratinocytes stimulated with TNF-α or LPS, exerted pro-differentiating activity, and reduced the expression of pro-inflammatory cytokines and antimicrobial peptides [171] (Table 1). These data support NPD-0614-13 and NPD-0614-24 as new therapeutic agents in the management of PS. However, further preclinical and clinical studies are required to evaluate the possible use of the aforementioned molecules for the treatment of PS and AD.

### 3.3. Exogenous AHR Ligands: Flavonoids and Indoles as a Phytochemical Dietary Source of AHR Ligands

Dietary AHR ligands are obtained mainly from foods with abundant L-Trp-derived indoles and polyphenols, such as vegetables and plant by-products [172,173]. These foods contain AHR ligands or pro-ligands. For example, indole-3-acetonitrile (I3ACN) and indole-3-carbinol (I3C) (also known as indole-3-methanol—I3M) are weak dietary AHR ligands but, in the acidic environment of the stomach, they undergo non-enzymatic condensation reactions that transform them into a variety of AHR ligands, including 2-(indol-3-ylmethyl)-3,3′-diindolylmethane, 3,3′-diindolylmethane (DIM), and ICZ [174]. Among these, ICZ shows the highest AHR agonistic activity [175,176]. The formation of relatively potent AHR ligands from precursors that have little or no AHR agonist activity is significant, especially considering that most dietary ligands are relatively weak AHR ligands. These dietary indoles and endogenous derivatives have an impact on the host immune defense capacity and homeostasis, especially in the control of bacterial gut colonization [156,177].

Both full and keratinocyte specific *Ahr*-deletion mouse lines show high TEWL, a clear parameter of defects in skin barrier integrity [107]. The removal of AHR ligands from the diet of control mice resembles defects of skin barrier integrity observed in mice with genetic deletion of *Ahr* in keratinocytes. On the other hand, the presence of I3C in the diet was sufficient to prevent the increased TEWL detected in ageing mice [107]. These results suggest that the regulation of skin barrier function through AHR is not exclusively due to the effect of UV-induced AHR ligands such as FICZ, and also indicate a systemic role for AHR ligand uptake from the diet. Interestingly, the antibiotic-mediated removal of microbiota prevents IMQ-induced skin inflammation through downregulation of Th17 immune response in conventional mice [178,179]. The relevance of specific microbiome-derived metabolites and AHR expression remains unexplored in the antibiotic-induced control of experimental PS.

Dietary AHR ligands I3C and DIM were compared to FICZ in the induction of Tregs and Th17 cells in a model of attenuated delayed-type hypersensitivity (DTH) [180] (Table 1). Both indoles decreased the induction of IL-17 but promoted IL-10 and FoxP3 expression in mice expressing AHR, attenuating skin inflammation. In contrast, FICZ exacerbated the DTH response and promoted Th17 cells, through activation of AHR [180]. Systemic administration of DIM also significantly alleviates skin erythema and edema in a mouse model of acute AD established using 2,4-dinitrofluorobenzene [181]. DIM promoted the differentiation of Treg cells and inhibited the Th2 and Th17 cells activation, but without significant effect on Th1 cells [181]. Finally, cutaneous delivery of [1-(4-chloro-3-nitrobenzenesulfonyl)-1H-indol-3-yl]-methanol, an I3C derivative, mitigates the onset of psoriasiform lesions by blocking MAPK/NF-κB/AP-1 activation [182], but the requirement of AHR vs. direct targeting of NF-κB signaling target is unclear.

The isomers indigo and indirubin are two indoxyl AHR agonists isolated from human urine [183]. In mammals, indoles produced by gut bacteria are absorbed by the host and circulate to the liver, where they are hydroxylated by CYP2E1 to form indoxyls and then sulphate—conjugated by sulphotransferases [184,185]. The salts of the resulting indoxyl sulfates are excreted in the urine. In vitro and in vivo studies with indirubin have reported its anti-inflammatory capacity [186,187]. Accordingly, this compound has been used in clinical trials for IBD treatment [188,189,190]. Additionally, since indigoids herbal remedies have been commonly used in traditional chinese medicine for treating dermatosis and skin lesions such as eczema, aphtha, or eruptions, a few clinical trials suggested that topical treatment with *Indigo naturalis* ointment is effective in treating PS [191,192] and AD [193]. Studies on cultured primary human keratinocytes indicated that the anti-psoriatic effects of *I. naturalis* extract rely on blocking keratinocyte proliferation, inducing keratinocyte differentiation, upregulating claudin-1 expression, and enhancing the function of tight junctions [194,195].

Polyphenols are another important group of phytochemical dietary products that interact with AHR. Polyphenols can be flavonoids and non-flavonoids. Curcumin and resveratrol are non-flavonoids that interact with AHR [196]. Some examples of flavonoid ligands are quercetin, kaempferol, apigenin, naringin, chrysin, diosmin, or tangeritin. Despite the similar chemical structures of various flavonoids, their role in controlling the activity of AHR can be very different and their reported effects as agonists or antagonists are sometimes contradictory. For example, many flavonoids have dual agonist–antagonist activity, depending on their concentration, in a species- or cell line-specific manner, by synergistic interactions with other ligands, or due to indirect activation of the AHR through inhibition of specific CyP and accumulation of another ligand [197,198,199,200]. Importantly, plant-based polyphenols are generally recognized as health-promoting, and some of them can be found as constituents of commercial nutraceutical formulations. Polyphenols show anti-inflammatory effects that have been extensively studied in IBD models [201,202,203,204,205]. Most polyphenols and flavonoids display antioxidant properties due to their chemical structure, which includes hydroxyl groups [206,207] that make them reducing agents and inhibitors of enzymes involved in ROS generation, like microsomal monooxygenases (acting directly on the enzyme or indirectly on other pathways or transcriptional regulation, e.g., by AHR antagonism) [200]. Moreover, some flavonoids modulate immune responses through AHR. For example, naringenin induces the generation of Treg cells [208] and quercetin induces tolerogenic LPS-matured DCs [209] by AHR-mediated pathways.

### 3.4. Microbial-Derived AHR Ligands

Indoles can also be generated through the microbial metabolism of L-Trp. Tryptophanase-positive commensal microbes from barrier organs (e.g., skin, digestive, or urinary tract) can use L-Trp as a nitrogen or energy source and metabolize it into indolic derivatives, some of which are AHR ligands or can be further transformed into higher affinity indoxyl sulfate ligands by hepatic host enzymes as explained above [184]. Among microbiota-derived indoxyl compounds, indoxyl-3-sulfate (I3S) is a potent ligand for human AHR, although it exhibits a lower affinity for mouse AHR [210]. Of note, I3S is undetectable in the blood of germ-free mice [211], indicating that its synthesis depends on commensal bacteria. I3S, and other metabolites derived from dietary L-Trp through modification by the microbiota, cross the blood–brain barrier and limit inflammation through AHR-driven mechanisms in astrocytes and microglia in EAE models [212,213].

Bacterial species that produce indoles through bacterial L-Trp metabolism and their effects on the host health have been described elsewhere [214,215]. Briefly, Lactobacilli species, especially *L. reuteri*, produce AHR ligands, e.g., indole-3-aldehyde (IAld) and indole-3-lactic acid (ILA). Additional microbial metabolites of dietary L-Trp like IAA, IAAld, tryptamine, or 3-methylindole (skatole) are also endowed with AHR-agonistic activity [216,217,218]. Intestinal IAld regulates IL-22 expression in ILC3 via AHR and this stimulates antimicrobial protein production, promoting resistance against pathogenic microorganism colonization (*Candida albicans*) and maintaining intestinal homeostasis [219]. Indoles produced by Lactobacilli also contribute to homeostasis-preventing IBD pathogenesis by inducing IL-22 in an AHR-dependent manner [220,221,222]. Furthermore, IAld and ILA can reprogram CD4+ intraepithelial lymphocytes (IELs) into CD4+ CD8αα+ double-positive IELs, which promote tolerance to dietary antigens [223].

L-Trp metabolites derived from the microbiota also play a regulatory function in the skin. A metagenomic study of skin microbiomes from control and AD individuals found that L-Trp metabolism pathways are attenuated in the skin microbiome of patients with AD [224]. Comparison of L-Trp metabolite levels between the skin of patients with AD and that of healthy subjects showed that IAId was the only metabolite significantly decreased on both lesional and non-lesional skin of patients with AD [225]. Furthermore, topical application of IAId alleviated skin inflammation in a mouse model of AD in an AHR-dependent fashion [225] (Table 1).

In addition to bacteria, yeast can also metabolize L-Trp into several potent indole AHR ligands. Eukaryotic microbiome residents on human skin are dominated by the *Malasseziacea**e* family [224]. *Malassezia* species produce ICZ, malassezin, indirubin, pityriacitrin, pityriazepin, and FICZ in L-L-Trp agar culture extracts and in skin samples from patients, but not in healthy controls [226,227,228,229]. *Malassezia* is a common skin-residing yeast that can become pathogenic in diverse skin diseases. *Malassezia* can cause pityriasis versicolor, also known as tinea versicolor, a superficial fungal infection of the skin characterized by the formation of hyperpigmented or hypopigmented plaques mainly in the back, chest, upper arms, and neck regions. While pityriasis may involve a high fungal load without significant inflammation, *Malassezia* yeasts are implicated in exacerbations of AD and seborrheic dermatitis, which are inherently inflammatory disorders. Indoles produced by *Malassezia* are AHR agonists that activate the classical AHR response genes, CYP1A1 and CYP1B1 [226,227,228,229]. However, they also exert cell-specific effects on the skin. Activation of the AHR pathway in immortalized human keratinocytes (HaCaT), infected with different strains of *Malassezia*, upregulated FLG, IVL, and transglutaminase [230], as well as IL-1β expression [231,232]. Furthermore, *Malassezia* increases the levels of Toll-like receptor 2 (TLR-2), IL-1β, IL-6, IL-8, cyclooxygenase 2 (COX-2), and MMP-9 expression in infected HaCaT cells [231,232]. Therefore, exacerbated AHR activation by *Malassezia*-derived agonists may result in inflammation and alteration of epidermal barrier function.

### 3.5. Toxicity of AHR-Exogenous Ligand TCDD in the Skin

Several environmental contaminants—such as TCDD and other PAHs (e.g., biphenyls, 7,12-dimethylbenz[a]anthracene (DMBA), methylcholanthrene, or benzo[a]pyrene (B[a]P))—are exogenous ligands of AHR. Exposure to dioxins, in particular TCDD, induces a cutaneous syndrome known as chloracne in humans, consisting of the development of multiple, small skin lesions (hamartoma), lasting for 2–5 years [16,233] (Table 1). Chloracne has been observed mainly in workers frequently exposed to high levels of dioxins. TCDD cutaneous toxicity causes an alteration in the differentiation and proliferation of the skin, resulting in the thickening of the interfollicular squamous epithelium, as well as metaplasia and hyperkeratinization of the ducts of the sebaceous gland, with comedone formation [16,233].

TCDD binding to AHR upregulates CYP1A1 and CYP1B1 mRNA and protein expression in keratinocytes [234]. The enzymatic activity of CYP1A1 generates reactive oxygen species (ROS) and can induce oxidative damage in the cells [18,101]. Because TCDD is structurally stable and poorly metabolized by CYP1A1, it leads to sustained AHR activation as well as to exacerbated ROS generation by CYP1A1 activity [18,233]. The deletion of CYP1B1 did not affect TCDD-induced ROS generation at least in endothelial cells, suggesting the main role for the AHR-CYP1A1 axis in TCDD-induced toxicity [18]. However, additional experiments in keratinocytes are required to understand the specific role of CYP1A1 vs. CYP1B1 enzymes in TCDD-induced skin alterations. FICZ also induces ROS generation in keratinocytes in an AHR-dependent fashion, driving the production of the proinflammatory cytokines IL-1α, IL-1β, and IL-6 [235]. AHR activation by TCDD in primary mouse keratinocytes upregulates the expression of the neutrophil-stimulating chemokine (C-X-C motif) ligand 5 (CXCL5) [236]. This effect was also observed in *Ahr*-sufficient mice exposed to UV light or after topical treatment with FICZ [236]. Thus, AHR is an important regulator of CXCL5-induced neutrophil recruitment, with implications for skin homeostasis and inflammation. Hence, TCDD shares molecular pathways similar to FICZ, but the skin alterations associated with TCDD exposure are due to its poor metabolism through CYP1A1, which exacerbates its effects.

In vitro studies using normal human epidermal keratinocytes demonstrate that TCDD accelerates differentiation and increases gene expression of several skin-barrier proteins, including FLG [17,237,238]. Using an organotypic coculture system, containing primary human fibroblasts and an immortalized human keratinocyte cell line, it was shown that TCDD specifically altered the keratinocyte differentiation program, without affecting proliferation and apoptosis [95]. Cotreatment with chemical or enzymatic antioxidants blocked TCDD-mediated acceleration of keratinocyte cornified envelope formation, an endpoint of terminal differentiation. Thus, TCDD-mediated ROS production is a critical step in the mechanism of accelerating keratinocyte differentiation [16]. Moreover, TCDD strongly increased IL-6 and IL-8 release in normal human epidermal keratinocytes [93].

CD4+ T cells from patients with AD and PS showed higher expression levels of AHR-related factors, such as AHR, CYP1A1, IL-17, and IL-22. In vitro treatment with TCDD of PBMCs and CD4+ T cells from patients with PS and AD showed upregulation of the aforementioned AHR-related genes. In contrast, FICZ inversely affected the differentiation of CD4+ T cells and their cytokine expression levels, as compared with TCDD [239]. These results suggest that environmental pollutants such as TCDD may contribute to the development or exacerbation of AD and PS.

### 3.6. Tapinarof—A Novel Treatment for PS and AD

Tapinarof (3,5-dihydroxy-4-isopropylstilbene), previously known as GSK2894512 or WBI-1001, is a naturally derived small molecule produced by bacterial symbionts of entomopathogenic nematodes [240,241]. It is structurally similar to the vegetal polyphenol resveratrol but differs significantly from its activity [242]. High throughput profiling studies revealed that tapinarof binds directly to AHR resulting in downregulation of inflammatory cytokines, including IL-17A, IL-17F, IL-19, IL-22, IL-23, and IL-1β in the IMQ-induced PS model [242]. Tapinarof also induces the expression of skin barrier genes related to keratinocyte differentiation in an AHR-dependent manner, including FLG and LOR [242,243] (Table 1). In fact, tapinarof displayed a pattern of biological responses reminiscent of that of the AHR agonist FICZ in the IMQ model [105].

Clinical studies have demonstrated that topical application of tapinarof is efficacious and well-tolerated in patients with AD and PS [244,245,246]. In addition to AHR, tapinarof interacts with Nrf2, cannabinoid receptor type 2, and monoamine oxidase B pathways [242]. Tapinarof displays intrinsic antioxidant activity through two phenol groups that scavenge ROS, and induces the AHR-Nrf2 transcription factor pathway, leading to the expression of antioxidant enzyme genes [242,247]. Despite its antioxidant activity, the therapeutic effects of tapinarof in the IMQ-mouse model of PS were not observed in *Ahr*-deficient mice, suggesting that tapinarof exerts its anti-inflammatory role in PS by controlling AHR signaling [242]. Tapinarof also inhibited T cell expansion and Th17-cell differentiation in vitro, reducing IL-17A and IL-17F secretion, which is relevant for PS treatment [242]. Furthermore, tapinarof treatment restores the downregulation of FLG and LOR expression induced by IL-4, a key cytokine in AD [243]. Finally, tapinarof induces AHR-mediated secretion of IL-24, which downregulates FLG and LOR expression and alters keratinocyte differentiation [243,248,249]. Hence, inhibition of the IL-24 signaling pathway might be considered to improve tapinarof therapeutic effects [243].

**Table 1 cells-10-03176-t001:** Role of direct AHR ligands and intermediate L-Trp-derived metabolites in psoriasis (PS) and atopic dermatitis (AD). Molecules that have been proved to induce AHR transcriptional activity are in black, while intermediate molecules of metabolic pathways are in grey.

Origin/Source	Molecule	Effects in PS or AD
L-Trp-derived metabolites of L-kynurenine pathway	L-Kynurenine (L-Kyn)	*Ido2*-deficient mice show exacerbated IMQ-induced PS [128]. Increase Kyn/Trp ratio in PS and AD patients [129,130].
	Kynurenic Acid (Kyn A)	Suppresses IL-23/IL-17 in vitro secretion in DC/CD4+ T cells, respectively, after LPS stimulation [134].
	Xanthurenic Acid	Not assessed in PS or AD.
	Cinnabarinic Acid	Not assessed in PS or AD.
	3-Hydroxyanthralinic acid	Proinflammatory role suggested in PS and AD. Induces the expression of CCL20 and IL-8 in keratinocytes in vitro [130].
	Quinolinic acid	Proinflammatory role suggested in PS and AD. Induces chemokines and adhesion molecules in keratinocytes and endothelial cells, respectively, in vitro [130].
	3-hydroxy-L-kynurenamine (3-HKA)	Protective role in IMQ-induced PS mouse model [133].
Serotonin pathway	5-Hydroxytryptophan (5(OH)Trp)	Controls IMQ-induced psoriasiform dermatitis [144].
Tryptamine pathway	Tryptamine	Not assessed in PS or AD.
Oxidative L-Trp metabolite	6-formylindolo [3,2-b]carbazole (FICZ)	Attenuates IMQ-psoriasiform skin inflammation by increasing FLG expression and reducing IL-17 and IL-22 levels [105]. Exacerbates the DTH response by promoting Th17 cells [180].
Synthetic	NPD-0614-13 NPD-0614-24	Protective role in three-dimensional models of psoriasis [171].
Dietary ligands	Indole-3-acetonitrile (I3ACN)	Not assessed in PS or AD.
	Indole-3-carbinol (I3C)	Controls IL-17 secretion and increases Foxp3 and IL-10 expression. Controls skin inflammation in a model of DTH [180].
	Indigo	Effective in the treatment of PS and AD patients [191,192,193]. Induce keratinocyte differentiation [194,195]
	Indirubin	Inhibits inflammatory reactions in DTH mouse model [187]
	3,3′-diindolylmethane (DIM)	Decreases IL-17 secretion while increasing Treg differentiation, thus controlling skin inflammation in a model of DTH [180]. Reduces Th2 and Th17 cell proliferation, increases Treg, attenuating atopic dermatitis-related immune responses [181].
Microbiota indole ligands	Indole-3-acetylaldehyde (IAAld)	Not assessed in PS or AD.
	Indole-3-acetic acid (IAA)	Not assessed in PS or AD.
	Indole-3-lactic acid (ILA)	Not assessed in PS or AD.
	Indole-3-aldehyde (IAld)	Metabolite significantly decreased on both lesional and non-lesional skin of patients with AD Alleviates skin inflammation in AD mouse model [225].
*Malassezia*	Malassezin Pityriacitrin Pityriazepin	Upregulate FLG and IVL genes in keratinocytes in vitro [229]. Induce proinflammatory mediators in keratinocytes in vitro [230,231]. Associated to exacerbated AD and seborrheic dermatitis.
Synthetic	2,3,7,8-tetrachlorodibenzo-*p*-dioxin (TCDD)	Induces chloracne syndrome in humans [12].
Bacterial symbionts of entomopathogenic nematodes	3,5-dihydroxy-4-isopropylstilbene (Tapinarof, GSK2894512 or WBI-1001)	Protective role in the IMQ-induced PS model, by downregulation of inflammatory cytokines, and improvement of skin-barrier function [242].

## 4. Concluding Remarks and Future Perspectives

The barrier function of the epidermis involves the correct expression and configuration of multiple components, including several proteins and lipids. When properly functioning, the epidermis layer prevents water loss and provides a barrier against the invasion of allergens and bacteria. Environmental factors (e.g., pollutants) and UV-derived Trp-related metabolites can trigger AHR expression in the epidermis. The controlled and autoregulated (through CyPs activity) AHR-mediated transcriptional expression favors epidermal renewal and barrier function. On the other hand, exacerbated and prolonged AHR activation, due to failure of regulatory mechanisms, increases ROS generation and induces secretion of proinflammatory mediators, leading to exacerbated keratinization response. AHR expression is also relevant for immune responses, including secretion of IL-10, IL-21, IL-17, and IL-22, for macrophage and DCs function, as well as for lymphocyte retention and survival in the skin. In addition, endogenous or microbial-mediated metabolism of L-Trp, either systemically or locally, can influence skin immune responses and barrier function through AHR activation. Hence, AHR is an important player in skin integrity and immunity in both homeostasis and disease. The development of AHR ligands that skew the homeostasis of the skin towards keratinocyte differentiation and curb immune responses (e.g., tapinarof, FICZ, or NPD-0614-13 and NPD-0614-24 compounds) is essential for the control of skin inflammation. Moreover, additional checkpoints in the metabolism of L-Trp, or its cellular uptake, emerge as important novel strategies to prevent cutaneous diseases, likely by regulating AHR activation.

There are important issues to consider nowadays and for the future regarding the biology of AHR and the regulation of its function in skin homeostasis and diseases. The classical ligands, FICZ and TCDD, promote AHR degradation, and the consequences of its depletion in the biological responses have not been evaluated, as well as the mechanism controlling reconstitution of AHR expression. Moreover, the studies aimed at targeting the endogenous generation of AHR specific ligands in immune cells and keratinocytes—mainly L-Trp-derived ligands and related metabolites—are scarce. There is an opportunity for metabolomics studies to rule out the role of AHR ligands as markers of disease progression or relapse. Importantly, the stoichiometry of all AHR ligands, as well as their effects induced under physiological levels, is mostly unknown. The future of AHR modulation should also include the potential TRM modulation, as well as the generation of Tr1 and Treg cells in PS and AD lesions. Finally, diet- or microbiota-modulation strategies to specifically increase AHR ligands generation should be explored, not only to prevent skin pathologies but also to control the systemic inflammation and co-morbid diseases of PS and AD patients.

## Figures and Tables

**Figure 1 cells-10-03176-f001:**
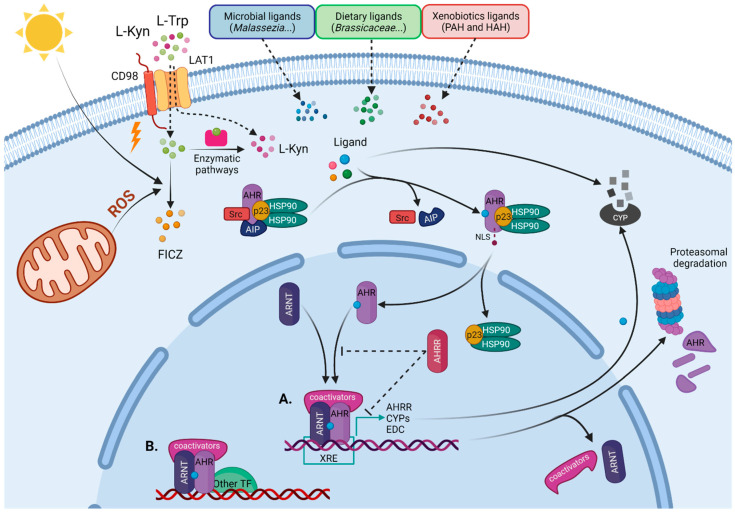
AHR genomic signaling pathway. Before ligand binding, AHR is bound by a chaperone complex (described in the text and the figure), which maintains its localization in the cytoplasm. Cells are exposed to different AHR ligands, such as bioproducts of microbiota, phytochemicals, xenobiotics, or endogenous ligands, mostly derived from L-tryptophan (L-Trp). When a ligand binds, the AHR changes its conformation and c-Src and AHR-interacting proteins (AIP) are released, resulting in the exposure of the nuclear localization signal (NLS) in the AHR’s N-terminus, that allows docking of importin β and mediates nucleocytoplasmic shuttling. Once in the nucleus, the ligand-activated AHR heterodimerizes with its protein partner, the AHR nuclear translocator (ARNT), at the time it dissociates of cytoplasmic chaperone complex. (**A**) The ligand–AHR–ARNT heterodimeric complex binds specific DNA sequences located in the promoter regions of target genes, named xenobiotic responsive elements (XRE), and recruits additional coactivators and components of the transcriptional machinery (described in the text) that are required to initiate transcription of the target gene. (**B**) The ligand–AHR–ARNT complex can also interact with non-canonical AHR partners and regulate additional target genes. (**A**) Canonical genes include enzymes of the cytochrome P450 (CyP) family and AHR repressor (AHRR). CyP enzymes metabolize AHR ligands and the AHRR competes with the AHR for interaction with the ARNT and DNA binding. After transcription, the AHR is exported out of the nucleus and is rapidly degraded by the proteasome. PAH—polycyclic aromatic hydrocarbon; HAH—halogenated aromatic hydrocarbon; ROS—reactive oxygen species; HSP90—90 kDa heat shock protein; EDC—epidermal differentiation complex; TF—transcriptional factor. Figure was created with BioRender.com.

**Figure 2 cells-10-03176-f002:**
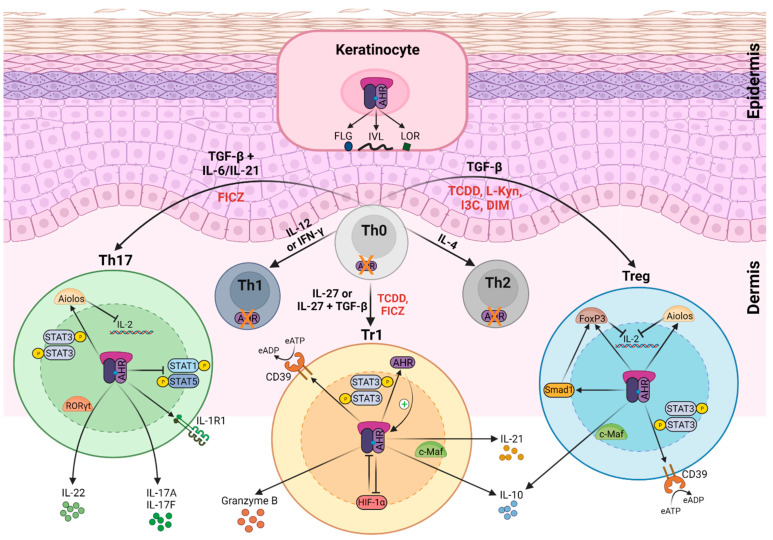
Effects of AHR signaling in keratinocytes and different T cell subsets. AHR expression depends on T helper (Th) cell subsets. In naïve CD4+ T cells (Th0), Th1 cells, and Th2 cells, AHR expression is negligible. Th17 express the highest AHR levels and regulatory T (Treg) cells and T regulatory type 1 (Tr1) cells show intermediate levels. AHR’s activation exerts multiple effects on T cells. Engagement by particular ligands—such as 2,3,7,8-tetrachlorodibenzo-*p*-dioxin (TCDD), 6-formylindolo [3,2-b]carbazole (FICZ), L-kynurenine (L-Kyn), indole-3-carbinol (I3C), or 3,3′-diindolylmethane (DIM) (indicated in red)—activate transcriptional programs, which regulate the effector functions of AHR-expressing T cell subsets. In Th17 cells, the AHR enhances interleukin (IL)-17A, IL-17F, and IL-22 release, in cooperation with RAR-related orphan receptor (ROR)γt. The AHR upregulates IL-1 receptor type 1 (IL-1R1) expression in Th17 cells. The AHR also inhibits signal transducer and activator of transcription (STAT)1 and STAT5, which negatively regulates the Th17 program, and together with STAT3, induced Aiolos expression that resulted in IL-2 silencing. In Tr1 cells, the AHR interacts with musculoaponeurotic fibrosarcoma (c-Maf) to induce the expression of IL-10 and IL-21, and with STAT3 to drive CD39 expression and its own expression, which acts as a positive feedback loop. The AHR also upregulates the expression of granzyme B and promotes hypoxia-inducible factor (HIF)-1α degradation. Similarly, the AHR induces IL-10 and CD39 in Treg cells, upregulates the expression of forkhead box (Fox)P3, and mothers against decapentaplegic homolog (Smad)1 and Aiolos. Smad1 controls the expression of FoxP3 and Aiolos cooperates with FoxP3 to repress IL-2 transcription. In keratinocytes, the AHR triggers the expression of genes of the epidermal differentiation complex (EDC) (described in the text) which encodes involucrin (IVL), loricrin (LOR), and filaggrin (FLG) proteins, among others. IFNγ—interferon gamma; TGF-β—transforming growth factor beta; CD39—cluster of differentiation 39; P—phosphorylation; eATP—extracellular adenosine triphosphate; eADP—extracellular adenosine diphosphate. Figure was created with BioRender.com.

**Figure 3 cells-10-03176-f003:**
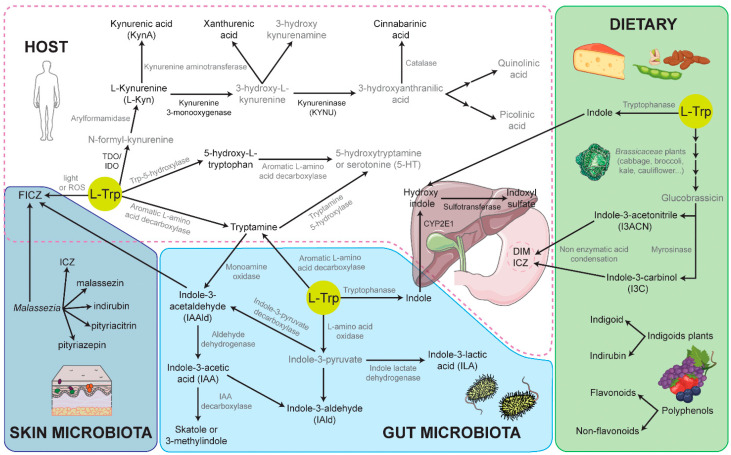
Summary of natural AHR agonist synthesis pathways in mammalian cells, associated microbiota, and dietary plants. HOST—endogenous L-tryptophan (L-Trp) metabolism via the L-kynurenine (L-Kyn), serotonin (5-HT), and tryptamine pathways; indoxyl synthesis in the liver from the diet- or microbial-derived indoles; gastric acid condensation of dietary indoles pro-ligands produces high-affinity indolic agonists; 6-formylindolo [3,2-b]carbazole (FICZ) is generated by L-Trp photo-oxidation, L-Trp chemical-oxidation, or by enzymatic deamination of tryptamine that yields the precursor indole-3-acetaldehyde (IAAld). SKIN MICROBIOTA—the AHR agonist, synthesized by skin-resident microbiota, such as *Malassezia* yeast species. GUT MICROBIOTA—indole-derived AHR agonists, provided by microbial metabolism of L-Trp, which can be further metabolized to indoxyl by the host. DIETARY—polyphenol- and indole (e.g., *Brassicaceae* plants)-rich foods provide exogenous ligands, either as direct AHR ligands or as pro-ligands that can be converted to AHR ligands by the host. L-Trp metabolic routes have been designed according to the KEGG Pathway Database (https://www.genome.jp/kegg/pathway/map/hsa00380.html (accessed on 8 October 2021) and PathBank (https://pathbank.org/view/SMP0000063 (accessed on 8 October 2021)). AHR ligands or enzymes regulated by AHR are indicated in black, and intermediate molecules and enzymes are in grey.

## Data Availability

L-Trp metabolic routes in Figure 3 have been designed according to the KEGG Pathway Database (https://www.genome.jp/kegg/pathway/map/hsa00380.html (accessed on 8 October 2021)) and PathBank (https://pathbank.org/view/SMP0000063 (accessed on 8 October 2021)).

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
