# Peer review of "Role of AHR Ligands in Skin Homeostasis and Cutaneous Inflammation"

_cells, 2021, doi:10.3390/cells10113176_

Round 1
Reviewer 1 Report
This is a review focussing on AHR ligands, skin and two skin diseases, which has a comprehensive reference list. It lacks novelty though and/or a tough approach to what is missing in research, and this would have been good. For instance - the questions of affinity, dose, off-target effects (such as antioxidant activities of many ligands) are not really untangled. Indeed, for most "ligands" it has never been shown formally that they bind to AHR. The stoichiometry of all these ligands in the body is usuallyl not addressed, which is a white elephant in the room.
Table 1 could be amended in this respect, or a second table or figure introduced.
The manuscript jumps a lot between humans and mouse models, especially regarding the PS and AD data and it would make the content of the paper much more accessible if this is clearly and everywhere distinguished.
Some comments in detail:
page 4 - In the sentence "AHR induces FoxP3 expression..." clarify if you mean induction via Ligand-binding/nuclear translocation/transcriptional activation"
page 4 - give a reference for last statement in paragraph two.
page 4, 4th paragraph - What is meant by "Finally, once gene transcription is finished..." Which mechanisms direct finishing, and add reference [66 not suitable]
page 5, 1st paragraph - it is not correct that FICZ and TCDD trigger differential responses in T cells. Gitta Stockinger has shown that in a paper where concentrations and route were similar.
page 5, 4th paragraph - can you add if AHR expression in M1 and M2 cells differs and how does it compare to LC?
page 6, ff - check text for unambiguity concerning full AHR-KO and conditional KO mice.
Figure 1 - curcumin in the diet is not an important AHR ligand, and only mentioned once in the text. Why not add important Trp sources in the diet? Also in the figure: Trp is not a source of glucobrasssicin, as suggested by the arrows. Please recheck the figure for correctness as figures are often important short-cuts for readers.
page 7 - The concept that Kyn down-stream metabolites are produced in inflammation while at the same time resulting in generation of Treg is noteworthy and it would help readers to better explain what might happen.
page 8 - there is a lot of jumping back and forth between humans and mice. Please change the text to make the "stories" more transparent.
page 9 - reference 166 is also ref 103
page 11, 3rd paragraph - a good example of listing many facts without a clear line of thought.
Concluding remarks - DNA damage is not really talked about in the text, so it is unclear why here. As concluding remarks go, it would be good to highlight the open questions.
Author Response
Reviewer 1:
Open Review
Comments and Suggestions for Authors
This is a review focussing on AHR ligands, skin and two skin diseases, which has a comprehensive reference list. It lacks novelty though and/or a tough approach to what is missing in research, and this would have been good. For instance - the questions of affinity, dose, off-target effects (such as antioxidant activities of many ligands) are not really untangled. Indeed, for most "ligands" it has never been shown formally that they bind to AHR. The stoichiometry of all these ligands in the body is usuallyl not addressed, which is a white elephant in the room.
We agree with the reviewer that information of ligands stoichiometry and their function under physiological concentration is scarce. We have indicated these aspects in the new “Concluding remarks and future perspectives” section. Please see page 16, lines 684-698.
Table 1 could be amended in this respect, or a second table or figure introduced.
We have indicated in the Table 1 and Figure 3 that the AHR-ligands in black and the intermediate metabolites in grey.
The manuscript jumps a lot between humans and mouse models, especially regarding the PS and AD data and it would make the content of the paper much more accessible if this is clearly and everywhere distinguished.
We have indicated in each case whether mice or human data are described.
Some comments in detail:
page 4 - In the sentence "AHR induces FoxP3 expression..." clarify if you mean induction via Ligand-binding/nuclear translocation/transcriptional activation".
We thank the reviewer by his/her comment. Evidences regarding AHR-mediated transactivation of FoxP3 has been indicated in the manuscript. Additional mechanisms of Treg regulation by AHR activation are also described. It is also added to the new Figure 2.
page 4 - give a reference for last statement in paragraph two.
The sentence has been rewritten and moved to page 6, lines 208-218. We have included the following references: Quintana, F.J.; et al. Nature 2008, 453, 65–71, doi:10.1038/nature06880; Duarte, J.H. et al. PLoS One 2013, 8, e79819, doi:10.1371/JOURNAL.PONE.0079819.
page 4, 4th paragraph - What is meant by "Finally, once gene transcription is finished..." Which mechanisms direct finishing, and add reference [66 not suitable]
Following reviewer suggestion, we have carefully described the mechanisms that negatively regulates AHR-mediated transcriptional activity. Reference 66 has been removed.
page 5, 1st paragraph - it is not correct that FICZ and TCDD trigger differential responses in T cells. Gitta Stockinger has shown that in a paper where concentrations and route were similar.
Following reviewer suggestion, we have included the published work of Brigitta Stockinger group regarding the effects of FICZ and TCDD in EAE model. Overall, these compelling data demonstrated that AHR activation can be modulated by multiple factors, including dose and scheme of administration.
page 5, 4th paragraph - can you add if AHR expression in M1 and M2 cells differs and how does it compare to LC?
We have carefully revised the bibliography and, for the best of our knowledge, this information is not described, so far.
page 6, ff - check text for unambiguity concerning full AHR-KO and conditional KO mice.
Following the reviewer suggestion, we have carefully revised the manuscript and clarified when the results were obtained using mice with conditional deletion of AHR in keratinocytes vs full KO mice.
Figure 1 - curcumin in the diet is not an important AHR ligand, and only mentioned once in the text. Why not add important Trp sources in the diet? Also in the figure: Trp is not a source of glucobrasssicin, as suggested by the arrows. Please recheck the figure for correctness as figures are often important short-cuts for readers.
Following the reviewer suggestion, we have designed a novel Figure 3, where the curcumin has been removed and diet Trp-sources are included. We also added examples of polyphenol rich food. L-tryptophan metabolic routes of new Figure 3 have been designed according to KEGG PATHWAY Database and PathBank.
Glucobrassicin (also indicated in the routes as Indolylmethyl glucosinolate) is a Trp-derived metabolite generated by the glucosinolate biosynthesis pathway as can be checked in this metabolic chart: https://www.genome.jp/pathway/map00966+C05837
and here https://pathbank.org/view/SMP0002445.
page 7 - The concept that Kyn down-stream metabolites are produced in inflammation while at the same time resulting in generation of Treg is noteworthy and it would help readers to better explain what might happen.
The concept that L-Kyn results in Treg generation and AHR underlie the regulation of autoimmunity and resolution of inflammation is included in the manuscript. Please see page 9, lines 343-357.
page 8 - there is a lot of jumping back and forth between humans and mice. Please change the text to make the "stories" more transparent.
We have indicated in each case whether mice or human data is described.
page 9 - reference 166 is also ref 103
We thank the reviewer by his/her comment. We have corrected the mistake.
page 11, 3rd paragraph - a good example of listing many facts without a clear line of thought.
We have rewritten this paragraph in order to better convey the message.
Concluding remarks - DNA damage is not really talked about in the text, so it is unclear why here. As concluding remarks go, it would be good to highlight the open questions
We have included in the last section a new paragraph of future perspectives. The DNA damage mention has been removed.
Reviewer 2 Report
Aryl hydrocarbon receptors (AHRs) regulate skin barrier function and have been implicated in immune and inflammatory responses to various skin conditions. This comprehensive and well-thought out review discusses AHR, its exogenous and endogenous ligands, and their role in skin homeostasis and disease, with a particular emphasis on atopic dermatitis and psoriasis. The table and figure provided are also very clear and informative, increasing the impact of the work. Overall, this study is suitable for publication with minor revisions. Comments for this manuscript are highlighted below.
Comments:
- In the conclusion, the authors briefly mention how AHR signaling could be exploited to develop novel strategies to prevent cutaneous disease. This concept should be expanded on either in the conclusion or throughout the text. This will increase the impact of the work, turning this review from a detailed report of what has already been done, to a roadmap for future directions in the field.
- Table 1 is very informative, however the legend is missing (please add) and some entries have different font, size, and color. Is there a reason for this? If so, please explain in the legend, otherwise make each entry consistent with the rest.
- A suggestion: the authors could compose another figure highlighting the role of AHR signaling in the immune response.
Author Response
Reviewer 2.
Open Review
Comments and Suggestions for Authors
Aryl hydrocarbon receptors (AHRs) regulate skin barrier function and have been implicated in immune and inflammatory responses to various skin conditions. This comprehensive and well-thought out review discusses AHR, its exogenous and endogenous ligands, and their role in skin homeostasis and disease, with a particular emphasis on atopic dermatitis and psoriasis. The table and figure provided are also very clear and informative, increasing the impact of the work. Overall, this study is suitable for publication with minor revisions. Comments for this manuscript are highlighted below.
We thank the reviewer for his/her positive appraisal of the manuscript.
Comments:
- In the conclusion, the authors briefly mention how AHR signaling could be exploited to develop novel strategies to prevent cutaneous disease. This concept should be expanded on either in the conclusion or throughout the text. This will increase the impact of the work, turning this review from a detailed report of what has already been done, to a roadmap for future directions in the field.
Following the reviewer suggestion, we have expanded the conclusion section highlighting the future perspectives in the field.
- Table 1 is very informative, however the legend is missing (please add) and some entries have different font, size, and color. Is there a reason for this? If so, please explain in the legend, otherwise make each entry consistent with the rest.
We thank the reviewer for his/her comments. The table has been replaced with a new one with the homogenous format and with the legend. We also added TCDD, tapinarof and NPD-0614-13/24 ligands in the table.
- A suggestion: the authors could compose another figure highlighting the role of AHR signaling in the immune response.
Following the reviewer suggestion, we have included new figure 2, highlighting AHR functions in immune cells in the skin.
Reviewer 3 Report
In this manuscript, Fernandez-Gallego et al. review the literature surrounding what is known about aryl hydrocarbon (AHR) signaling in the skin, particularly in inflammation.
The manuscript is well-written and presents a compelling case for the role of AhR signaling (endogenous and exogenous) in the pathology and, potentially, the pharmacology of inflammatory skin diseases like AD and psoriasis. The authors cover a lot of ground here and do so fairly comprehensively.
Specifics points for the authors to address below:
In the introduction the authors mention that compromised skin barrier integrity can lead to xerosis. It would be a good idea here to briefly introduce the concept of transepidermal water loss, as it is critical to these processes. It may also be worth mentioning that loss of barrier integrity can lead to colonization with microorganisms, penetration by allergens, etc.
In section 1.1. Authors mention that AHR activation by ligands can trigger widely varying effects depending on cell type, ligand concentration, interaction with EGFR etc. Authors should probably mention species here as well; since a good amount of the literature examining the effects of AhR ligands on skin-related processes has been performed in mouse (as the authors discuss in this manuscript), it is appropriate to mention that AhR in human and mouse has low sequence homology at the C-terminal transactivation domain, potentially leading to divergent signaling paradigms between these species (see the Flaveny et al papers “Differential gene regulation by the human and mouse aryl hydrocarbon receptor” [2010] and “The mouse and human Ah receptor differ in recognition of LXXLL motifs.” [2008]. )
It might be useful for the reader unfamiliar with AHR signaling broadly if the authors made a mechanistic figure showing canonical AHR signaling including nuclear localization, binding cofactors, etc, so that readers can see visually where AhR is bound to what other proteins, etc.
“Upon agonist binding…” this sentence makes it sound like AhR enters the nucleus before binding ARNT, but ARNT stands for “AhR nuclear translocator.” Is this a misnomer or does AhR actually bind ARNT in the cytoplasm? Questions like this would be best answered with the figure I mentioned above.
“In keratinocytes, AHR triggers….S100A proteins…such as S100A7.” It would be good for authors to mention that S100A7 is also known as psoriasin, given its importance to psoriasis and the authors’ focus on inflammatory skin diseases. Readers will likely remember this point when they are reading further into the manuscript.
“AHRR inhibits AHR signaling in two ways. First AHRR competes with AHR for interaction with ARNT and XRE binding which, in turn, decreases gene expression. Likewise, HIF1A competes with AHR for its interaction with ARNT. Second…” Is there any relationship between AHRR and the competitive binding of HIF-1A to HIF-1B? Or are the authors just mentioning HIF because its another method of downregulating AHR signaling?
Authors state that xxx% of L-Trp goes to different metabolic pathways in mammalian cells, microbes, and plants. It seems obvious, but authors should also mention, in addition to the pathways by which L-Trp is metabolized, that it is also incorporated into proteins directly as an amino acid.
Authors discuss tryptophan being catabolized to kynurenine. Since both of these molecules are chiral, it’s probably best to refer to them as L-Trp and L-Kyn, as these are the product and substrate of the reactions being discussed by the authors, at least in mammals. This may seem pedantic, but it has been reported that the D enantiomer of kynurenine doesn’t seem to have the same effects as the L enantiomer, as demonstrated by lack of effects on fibroblasts at the same concentration of ligand (see PMID: 24637853 for one example), and L amino acids are the form relevant to humans.
Minor point, but the authors have omitted a few of the enzymes in Figure 1 (for example, after L-Trp is metabolized by IDO, it is then metabolized by aryl formamidase before it becomes L-kynurenine).
“When IMQ-induced PS-like dermatitis was assessed in Ido2-deficient mice…” Up until this point authors have only referred to Indoleamine-2,3-dioxygenase as IDO. It might be worthwhile then, before this point, to explain the relevant differences between IDO1 and IDO2.
The paragraph beginning “when IMQ-induced PS-like dermatitis was assessed in IDO2-deficient mice” is dense and a little bit hard to follow. Authors bring up lots of associations of the levels of tryptophan metabolities and several pathologies including PSO (and PASI specifically) as well as cancer cells, and inflammatory diseases, and mentioning keratinocytes and cancer cells and immune infiltrating cells etc. If authors could streamline this section a little bit and make it flow more logically and explain what are the repercussions of the things that they are saying (ex: these have higher levels of this, suggesting that…, as opposed to…). Just try and make this section more logical and read more easily.
“This pathway could play a relevant role in the skin of vitiligo patients.” What is the evidence for this? If authors are going to mention this at all they should briefly explain and not simply provide a reference.
Authors mention NPD-0614-13/24 and some data in primary cells. Are there any preclinical animal studies or clinical investigation of these compounds yet?
The authors need not address or rebut this next point, but I just wanted to mention something: The authors make a compelling case for the anti-inflammatory properties of many AhR ligands in skin conditions. I’m curious now, because there is a fair amount of literature now demonstrating anti-scarring effects of kynurenine and kynurenic acid via AhR in fibroblasts in the dermis (to give just a couple examples, PMID: 24637853, 23877570, 27144507, 26992058, 30027295), to the point where kynurenic acid-based drugs (fibrostop, FS2) have been developed for topical application for hypertrophic scar and keloid (see “The safety and tolerability of topically delivered kynurenic acid in humans. A phase 1 randomized double-blind clinical trial”. [2018] J. Pharm. Sci. and “A randomized, double-blind, activate-and placebo-controlled trial evaluating a novel topical treatment for keloid scars.” [2021] J. Drugs in Dermatol.) I believe that this compound has progressed to phase II trials as well, but am not aware of any other publications (yet). The literature surrounding kynurenine derivatives and fibrosis has pretty much completely revolved around roles in the dermis, especially on fibrobolasts but, if it’s being applied topically (as it has been for many animal models and in the human trials) then it’s fair to assume that there might be epidermal effects as well, particularly since epidermal inflammatory signaling is known to positively contribute to development of dermal fibrosis, particularly in the cases of hypertrophic scar and (perhaps especially) keloids. Obviously a rigorous discussion of this concept is outside of the scope of this manuscript, but I just wanted to mention it in case the authors wish to comment on this in the manuscript or, at the very least, hopefully find it interesting and maybe something to explore in their future research. After all, topically, kynurenine must pass through the epidermis to get to the dermis eventually, so it seems reasonable to think that it might have effects on the epidermis as well in this context, but it seems that much of the fibrotic research of tryptophan derivative compounds has ignored this possibility. Just something to think about!
Authors state that TCDD activates AHR and upregulates CYP1A1 and CYP1B1 expression in keratinocytes, but then that TCDD-AHR signaling induces activity of CYP1A1 but not CYP1B1. Does the upregulation of gene expression of CYP1B1 just not translate to more protein? Or does more protein not contribute to more metabolism?
IL-1A and 1B should have alpha and beta instead of A and B since authors are referring to the cytokines and not the genes. (in line 510)
“Associated with TCDD expositions” (line 516) authors should replace “expositions” with “exposure.”
Line 520” “in an organotypic coculture system.” What were the keratinocytes co-cultured with?
Author Response
Reviewer 3:
Open Review
Comments and Suggestions for Authors
In this manuscript, Fernandez-Gallego et al. review the literature surrounding what is known about aryl hydrocarbon (AHR) signaling in the skin, particularly in inflammation.
The manuscript is well-written and presents a compelling case for the role of AhR signaling (endogenous and exogenous) in the pathology and, potentially, the pharmacology of inflammatory skin diseases like AD and psoriasis. The authors cover a lot of ground here and do so fairly comprehensively.
We thank the reviewer for his/her positive appraisal of the manuscript
Specifics points for the authors to address below:
In the introduction the authors mention that compromised skin barrier integrity can lead to xerosis. It would be a good idea here to briefly introduce the concept of transepidermal water loss, as it is critical to these processes. It may also be worth mentioning that loss of barrier integrity can lead to colonization with microorganisms, penetration by allergens, etc.
Following reviewer suggestion, we have described the concept of transepidermal water loss and its relevance for AD. Please see page 2, lines 46-49.
In section 1.1. Authors mention that AHR activation by ligands can trigger widely varying effects depending on cell type, ligand concentration, interaction with EGFR etc. Authors should probably mention species here as well; since a good amount of the literature examining the effects of AhR ligands on skin-related processes has been performed in mouse (as the authors discuss in this manuscript), it is appropriate to mention that AhR in human and mouse has low sequence homology at the C-terminal transactivation domain, potentially leading to divergent signaling paradigms between these species (see the Flaveny et al papers “Differential gene regulation by the human and mouse aryl hydrocarbon receptor” [2010] and “The mouse and human Ah receptor differ in recognition of LXXLL motifs.” [2008]. )
We thank the reviewer for his/her suggestion. we have mentioned interspecies differences in the list of factors that influenced AHR function and included a paragraph to explain and discuss implication of human and mouse AHR differences. Please see page 2, lines 78-89.
It might be useful for the reader unfamiliar with AHR signaling broadly if the authors made a mechanistic figure showing canonical AHR signaling including nuclear localization, binding cofactors, etc, so that readers can see visually where AhR is bound to what other proteins, etc.
Following the reviewer suggestion, we have included a new figure 1, showing the canonical AHR signaling pathway.
“Upon agonist binding…” this sentence makes it sound like AhR enters the nucleus before binding ARNT, but ARNT stands for “AhR nuclear translocator.” Is this a misnomer or does AhR actually bind ARNT in the cytoplasm? Questions like this would be best answered with the figure I mentioned above.
We thank the reviewer for his/her suggestion. ARNT structurally is very similar to AHR but one key difference between them is that ARNT only contains a NLS in the bHLH domain for constitutive nuclear localization and AHR contains both nuclear localization and nuclear export signals. It means that after protein translation in cytoplasm by ribosomes, ARNT is imported to the nucleus where it carries out its function interacting as nuclear factor with AHR, AHRR or HIF1α. As indicated above, we have included a new figure 1 that will help readers to understand the mechanism of AHR activation.
“In keratinocytes, AHR triggers….S100A proteins…such as S100A7.” It would be good for authors to mention that S100A7 is also known as psoriasin, given its importance to psoriasis and the authors’ focus on inflammatory skin diseases. Readers will likely remember this point when they are reading further into the manuscript.
Following reviewer suggestion, we have indicated in line 268 of page 7 psoriasin as the alternative name of S100A7.
“AHRR inhibits AHR signaling in two ways. First AHRR competes with AHR for interaction with ARNT and XRE binding which, in turn, decreases gene expression. Likewise, HIF1A competes with AHR for its interaction with ARNT. Second…” Is there any relationship between AHRR and the competitive binding of HIF-1A to HIF-1B? Or are the authors just mentioning HIF because its another method of downregulating AHR signaling?
We have clarified this aspect in the manuscript. Please see lines 163-164 in page 4.The AHR transcriptional activity is affected by HIF1α interaction with ARNT, which is independent of AHRR-mediated inhibition.
Authors state that xxx% of L-Trp goes to different metabolic pathways in mammalian cells, microbes, and plants. It seems obvious, but authors should also mention, in addition to the pathways by which L-Trp is metabolized, that it is also incorporated into proteins directly as an amino acid.
Following the reviewer suggestion, we have mentioned the relevance of L-Trp for protein synthesis in page 7, lines 310-311.
Authors discuss tryptophan being catabolized to kynurenine. Since both of these molecules are chiral, it’s probably best to refer to them as L-Trp and L-Kyn, as these are the product and substrate of the reactions being discussed by the authors, at least in mammals. This may seem pedantic, but it has been reported that the D enantiomer of kynurenine doesn’t seem to have the same effects as the L enantiomer, as demonstrated by lack of effects on fibroblasts at the same concentration of ligand (see PMID: 24637853 for one example), and L amino acids are the form relevant to humans.
Following the reviewer suggestion, we have replaced Trp and Kyn for L-Trp and L-Kyn, respectively, in full text, figures and table.
Minor point, but the authors have omitted a few of the enzymes in Figure 1 (for example, after L-Trp is metabolized by IDO, it is then metabolized by aryl formamidase before it becomes L-kynurenine).
We thank the reviewer by his/her comment. We have revised L-tryptophan metabolic routes using KEGG PATHWAY Database (https://www.genome.jp/kegg/pathway.html > https://www.genome.jp/kegg/pathway/map/hsa00380.html) and PathBank (https://pathbank.org/ > https://pathbank.org/view/SMP0000063) and included in the figure the omitted enzymes, among them aryl formamidase enzyme and N-formyl-kynurenine intermediate metabolite.
“When IMQ-induced PS-like dermatitis was assessed in Ido2-deficient mice…” Up until this point authors have only referred to Indoleamine-2,3-dioxygenase as IDO. It might be worthwhile then, before this point, to explain the relevant differences between IDO1 and IDO2.
Following the reviewer suggestion, detailed information about IDO1 and IDO2 are now included in the manuscript. Please see pages 8-9, lines 327-343.
The paragraph beginning “when IMQ-induced PS-like dermatitis was assessed in IDO2-deficient mice” is dense and a little bit hard to follow. Authors bring up lots of associations of the levels of tryptophan metabolities and several pathologies including PSO (and PASI specifically) as well as cancer cells, and inflammatory diseases, and mentioning keratinocytes and cancer cells and immune infiltrating cells etc. If authors could streamline this section a little bit and make it flow more logically and explain what are the repercussions of the things that they are saying (ex: these have higher levels of this, suggesting that…, as opposed to…). Just try and make this section more logical and read more easily.
We thank the reviewer for his/her comments. We have rewritten this section.
“This pathway could play a relevant role in the skin of vitiligo patients.” What is the evidence for this? If authors are going to mention this at all they should briefly explain and not simply provide a reference.
We have removed the sentence regarding vitiligo, to better focus the revision on PS and AD.
Authors mention NPD-0614-13/24 and some data in primary cells. Are there any preclinical animal studies or clinical investigation of these compounds yet?
The description of NPD-0614-13/24 ligands have been performed in in vitro skin diseases models very recently, and further preclinical and clinical studies are required to evaluate the possible use of the aforementioned molecules for the treatment of PS and AD. This information has been included in the manuscript. Please see pages 11-12, lines 455-464.
The authors need not address or rebut this next point, but I just wanted to mention something: The authors make a compelling case for the anti-inflammatory properties of many AhR ligands in skin conditions. I’m curious now, because there is a fair amount of literature now demonstrating anti-scarring effects of kynurenine and kynurenic acid via AhR in fibroblasts in the dermis (to give just a couple examples, PMID: 24637853, 23877570, 27144507, 26992058, 30027295), to the point where kynurenic acid-based drugs (fibrostop, FS2) have been developed for topical application for hypertrophic scar and keloid (see “The safety and tolerability of topically delivered kynurenic acid in humans. A phase 1 randomized double-blind clinical trial”. [2018] J. Pharm. Sci. and “A randomized, double-blind, activate-and placebo-controlled trial evaluating a novel topical treatment for keloid scars.” [2021] J. Drugs in Dermatol.) I believe that this compound has progressed to phase II trials as well, but am not aware of any other publications (yet). The literature surrounding kynurenine derivatives and fibrosis has pretty much completely revolved around roles in the dermis, especially on fibrobolasts but, if it’s being applied topically (as it has been for many animal models and in the human trials) then it’s fair to assume that there might be epidermal effects as well, particularly since epidermal inflammatory signaling is known to positively contribute to development of dermal fibrosis, particularly in the cases of hypertrophic scar and (perhaps especially) keloids. Obviously a rigorous discussion of this concept is outside of the scope of this manuscript, but I just wanted to mention it in case the authors wish to comment on this in the manuscript or, at the very least, hopefully find it interesting and maybe something to explore in their future research. After all, topically, kynurenine must pass through the epidermis to get to the dermis eventually, so it seems reasonable to think that it might have effects on the epidermis as well in this context, but it seems that much of the fibrotic research of tryptophan derivative compounds has ignored this possibility. Just something to think about!
We thank to the author for bring our attention to this issue. It is certainly something to think about. We have briefly mentioned this topic in the manuscript. Please see page 9-10, lines 390-403.
Authors state that TCDD activates AHR and upregulates CYP1A1 and CYP1B1 expression in keratinocytes, but then that TCDD-AHR signaling induces activity of CYP1A1 but not CYP1B1. Does the upregulation of gene expression of CYP1B1 just not translate to more protein? Or does more protein not contribute to more metabolism?
We have clarified these issues in the manuscript. TCDD increase expression of CYP1A1/CYP1B1 mRNA and protein expression. However, CYP1A1 play a major role in the TCDD-induced ROS generation.
IL-1A and 1B should have alpha and beta instead of A and B since authors are referring to the cytokines and not the genes. (in line 510)
Thank for the comment. The text has been amended.
“Associated with TCDD expositions” (line 516) authors should replace “expositions” with “exposure.”
Thank for the comment. The text has been amended.
Line 520” “in an organotypic coculture system.” What were the keratinocytes co-cultured with?
Following the reviewer suggestion, the details of the study were included in the manuscript. Please see page 15, lines 621-622.
Round 2
Reviewer 3 Report
I appreciate the authors' attention to detail and their engagement with my comments and with those of the other reviewers. In my opinion, in its current form, the manuscript is clearly written, interesting, and an impressive synthesis of the work of the authors and other groups seeking to investigate the roles of AhR signaling in inflammatory skin diseases. I am happy to recommend the manuscript in its current form for publication with no further need for revisions. I'd like to thank the authors for a good read, and I look forward to seeing future work in this field!